# REPRESENTATION LEARNING FOR SEISMIC HAWKES PROCESSES

**David Belanger, Yaniv Ovadia, & Maxwell Bileschi**
Google Brain
{dbelanger,yovadia,mlbileschi}@google.com

**Brendan Meade** *
Harvard University
meade@fas.harvard.edu

## ABSTRACT

Today, more than half a billion people live under the threat of devastating earthquakes. The 21st century has already seen earthquakes kill more than 800,000 people and cause more than 300 billion in damage. Despite these impacts and decades worth of research into the physics of seismic events, existing earthquake predictions are often too inaccurate to be useful for issuing actionable warnings. It is possible that deep learning could help close this gap, but as researchers we must proceed with care. First, there is a limited supply of historical data, and thus overfitting is a key concern. Second, it is important that models' predictions and parameters are interpretable, so that they can be used to generate and validate hypotheses about the underlying physical process. In response, we provide a case study of applying deep learning to forecasting seismic events in Southern California. We replace small components of a popular Hawkes process model for earthquake forecasting with black-box neural networks, with the goal of maintaining a similar level of interpretability as the original model. Using experiments on about three decades of earthquake hypocenter and magnitude estimates, we visualize our learned representations for earthquake events and discuss interpretability-accuracy tradeoffs. Our visualization may be useful to provide refinements to the Utsu/Ohmori law for the time-decay of aftershock productivity (Utsu, 1971).

## 1 INTRODUCTION

Natural science researchers often employ idealized statistical models that can be reliably fit using limited observational data and have a small number of interpretable parameters. Unfortunately, these criteria are not satisfied by many contemporary deep learning models. It should be a key priority of our community to overcome these barriers, as representation learning may significantly accelerate scientific progress by providing novel methods for understanding complex processes.

This paper describes preliminary steps in this direction for earthquake science. We begin with a popular spatio-temporal Poisson process model (Sec. 3) and recast it as a convolutional neural network (CNN), discretizing in both time and space (Sec. 4). This formulation suggests a variety of simple neural network extensions to the model (Sec. 5) that introduce limited additional parameters, and for which the induced representations have natural interpretations. Then, we study our models using decades of earthquake data from Southern California and visualize the learned representations (Sec. 6). Our methods may be of general interest for modeling point process data.

## 2 EARTHQUAKE FORECASTING

Each earthquake event we observe is characterized by a location (longitude, latitude, depth), a time $t$, and a magnitude $M$. We define a *target event* as any event with magnitude above a threshold $M_{\text{target}}$ and $N_{x,y,t}$ as the number of target events occurring in a given space-time interval. Our goal is to model $P(N_{x,y,t}|H_t)$, where $H_t$ is the history of events before $t$. Crucially, the model is allowed to condition on the history of *all* events, including those with magnitude below $M_{\text{target}}$. Such modeling is key for studying the dynamics of *foreshocks*, small-magnitude events directly preceding catastrophic events (Dodge et al., 1995).

---

*Work done while a visiting researcher at Google.

Earthquakes can be forecast over many timescales. Early warning systems seek to provide seconds to minutes of warning depending on whether or not analysis of a fast propagating P-wave suggests an event is a catastrophic rupture (Allen & Kanamori, 2003). Medium-term forecasts for the next week or month have been proposed for issuing evacuation warnings (Keilis-Borok & Kossobokov, 1990). Long-term forecasts over decades are useful for setting insurance rates, urban planning, and anticipating emergency responses (Bird et al., 2015). We focus on medium-term forecasting, where we seek to model the spatio-temporal dynamics of bursts of intense activity.

## 3   EARTHQUAKE SCALING LAWS AND THE ETAS MODEL

A collection of empirical scaling laws have been shown to approximately describe the behavior of many seismically-active regions around the world. Let $N(M)$ be the overall number of earthquakes with magnitude greater than $M$ for a given fault system. The Gutenberg-Richter law states that $N(m) \propto \exp(-bM)$, with $b \approx 1$ (Gutenberg & Richter, 1944). Next, let $A(t)$ be the total number of aftershocks until time $t$ has elapsed since a large-magnitude event. The Ohmori law dictates that $A(t) \propto t^{-p}$, with $p \approx 1$ (Omori, 1894). Finally, let $A(M)$ be the number of aftershocks produced by a mainshock of magnitude $M$. The Utsu/Ohmori law specifies $A(M) \propto \exp(\alpha m)$ (Utsu, 1971).

These empirical laws earthquake systems are consistent with the idea that earthquake sequences may be self-exciting, and thus it may be promising to model them using a Hawkes process (Hawkes, 1971). Let $\{t_1, \ldots, t_n\}$ be the occurrence times of observed events. A Hawkes process is a Poisson process where the time-dependent intensity $\lambda(t)$ at $t > t_n$ is given by $\mu + c \sum_{i=1}^{n} \exp(\frac{1}{\tau}(t - t_i))$, with $c > 0$. For certain values of the parameters, draws from such a process will contain bursts of events separated by periods of quiescence.

The *epidemic type aftershock sequences* (ETAS) model (Ogata, 1988) is a popular extension of the Hawkes process for spatio-temporal earthquake data. Its parametrization is specifically designed such that the system's behavior is consistent with the above scaling laws. Let $\lambda(\text{longitude}, \text{latitude}, t)$ be the intensity of a Poisson process over target events. We have,

$$\lambda(x, y, t) = \mu(x, y) + c \sum_{(x_i, y_i, t_i, m_i) \in H_t} \nu_i(x, y, t). \tag{1}$$

Here, $\mu(x, y)$ is a time-independent background seismicity rate and $\nu_i > 0$ is:

$$\nu_i(x, y, t) = (t - t_i)^{-p} \exp(\beta \sqrt{(x - x_i)^2 + (y - y_i)^2}) \exp(\alpha m_i). \tag{2}$$

Note that $(p, \beta, \alpha)$, can be interpreted as characteristic length scales for time, space, and magnitude.

## 4   DISCRETIZED ETAS AS A CONVOLUTIONAL NETWORK

The ETAS model is commonly parameterized for continuous time and space, but it can be naturally adapted to data with discretized positions and locations. Here, the total number of target events in a given space-time grid cell is Poisson distributed with a per-grid-cell rate that depends on the history of nearby events (Helmstetter et al., 2006). This approach is attractive because evaluation of (1) on a dense grid can be parallelized using optimized code for convolutions, similar to WaveNet (Van Den Oord et al., 2016), though it precludes using time-varying $p$ and $\beta$ (Helmstetter et al., 2006). We first construct a 3-dimensional grid of observations, where each cell's value is the sum of $\exp(\alpha m_i)$, where $i$ indexes the events occurring in the cell. Then, we evaluate (1) at all $(x, y, t)$ using a stack of separable convolutions across the time and space. Note that the convolutional kernels are controlled completely by scalar parameters $p$ and $\beta$. Error in approximating (1) using a fixed-size convolution is minor because $\nu_i$ decays quickly with distance in time and space.

## 5   NEURAL NETWORK EXTENSIONS TO ETAS

The discretized ETAS model employs a structured, compactly-parametrized convolution applied to a learned per-grid-cell representation. This representation, the total exponentiated magnitude of events, provides useful visualization and interpretation, since its influence on the prediction is linear.

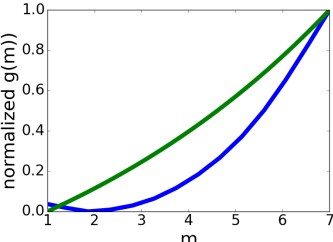

Figure 1: Learned $g(m)$ for MLP-ETAS (blue) vs. ETAS (green).

| Model | Unc | D-ETAS | MLP-ETAS |
|---|---|---|---|
| Gain | 0 | 176 | 185 |
| Model | Linear | SL-DCNN | DCNN |
| Gain | 290 | 363 | 414 |

Table 1: Likelihood gain of various models (higher is better).

With this restatement of the ETAS model as a CNN, it is clear that there are a variety of opportunities to generalize this model using learned neural network layers. Going forward, boldface will be used for the names of models that will be compared in our experiments.

In the discretized ETAS model (**D-ETAS**) above, each event is represented by the scalar $g(m) = \exp(\alpha m)$. As an alternative, we introduce **MLP-ETAS**, which uses a multi-layer perceptron with ReLU non-linearities for $g(\cdot)$. This provides a data-driven method for learning an alternative to the Utsu//Ohmori law that can be visualized easily (Fig. 1). In future work, $g(\cdot)$ could also take per-event covariates, such as depth, as inputs. **MLP-ETAS** is similar to the *pointnet* of Qi et al. (2017), where point clouds are represented by pooling per-point representations produced by an MLP.

Next, we introduce the **SL-DCNN** model. Here, $\lambda$ is produced by a multi-layer stack of convolutions and ReLU transformations. Unlike a generic CNN, however, the convolutions are parametrized using the scaling law approach described in the previous section. Unlike ETAS, we maintain multiple feature channels at every layer. These are combined using a local fully-connected layer before each ReLU. Parametrizing convolutions this way may be useful in other spatio-temporal modeling problems with scarce training data, especially where we seek to associate features with interpretable length-scales. In future work, it would be interesting to learn separate length-scales for each channel.

# 6 EXPERIMENTS

We fit models to the comprehensive catalog of events in Southern California, with magnitudes ranging from imperceptible to catastrophic. We use 1981-1995 for training and 1995-2004 for validation. As this is preliminary work, all results are presented on our validation data, on which we tune our hyperparameters. We use $M_{\text{target}} = 3.5$ and a grid cell size of 5 km $\times$ 7 km $\times$ 1 week.

In developing our ETAS extensions, our goal is not to achieve the best possible predictive performance, but instead to fit the data well using a model with interpretable representations and parameters. However, it is important to understand what performance this sacrifices. Therefore, we consider **Linear**, a Poisson regression model with hand-designed non-local features, such as the spatial and temporal distances from the most recent target event, and **DCNN** a multi-layer CNN, with generic 3-dimensional convolutions that we carefully regularize to avoid overfitting. Finally, we have the **Unc** model, where the Poisson rate consists entirely of the $\mu(x, y)$ term in (1). In future work, it would be natural to compare to the *neural Hawkes process* (Mei & Eisner, 2017). Fitting this to our data is challenging, though, as likelihood computation cannot be parallelized across time and space.

In Table 1, we evaluate models in terms of *likelihood gain*, their increase in log-likelihood vs. the **Unc** model. This is a standard metric for seismic forecasting. We find that **MLP-ETAS** is comparable to **D-ETAS**. However, the deep models are considerably better. This suggests that the ETAS family of models is insufficient and that we should develop new models that are both interpretable and close this performance gap. In Fig. 1, we plot the learned $g(m)$ function for **MLP-ETAS** vs. **D-ETAS** . Curves are normalized such that $g(\cdot)$ is in $[0, 1]$, as any scale factor can be absorbed into $c$ in (1). We find that our learned MLP produces a qualitatively different curve, which may be useful for proposing refinements to the Utsu/Ohmori law. Of course, further work is needed to ensure that estimation of this curve is reliable, and that the shape of the curve can be used to inform hypotheses about alternative scaling laws.

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
