# OpenReview forum: "Representation Learning for Seismic Hawkes Processes"
_ICLR.cc/2018/Workshop — Reject_

### Official Review · AnonReviewer1 · 2018-03-10
**Unclear if neural networks are a great choice for this application**

**Rating:** 5
**Confidence:** 3

**Review:**

This short three page paper essentially proposes to use neural networks as part of a standard model for forecasting seismic events.  The application is certainly consequential and the quality of the writing is generally good.  However, there isn't much evidence that neural networks are a great choice for this application.  There is an impression of the motivation simply being that neural networks have had good performance on a number of various tasks, so it's worth trying them for this task as well, which seems a bit too arbitrary for presentation.  For example, is a MLP really needed to learn something as simple as the blue curve in Figure 1?  Such structure can be easily learned with very standard interpretable models.  I would be concerned in this case that MLPs could overfit, and also you would lack a natural framework for making uncertainty estimates, which seem critically important in this setting.  If there had been more empirical evidence relative to good models, or careful motivation about why specifically neural networks are a good choice, I may have been swayed.  But as it stands with the current presentation, the application of MLPs here seems a bit too arbitrary.

Minor: From the abstract: "We replace small components of a popular Hawkes process model for earthquake forecasting with black-box neural networks, with the goal of maintaining a similar level of interpretability as the original model."  It would  be better to be specific about what these "small components" are in the abstract.

---

### Official Review · AnonReviewer3 · 2018-03-11
**Interesting application, not sure the exposition is quite thorough enough yet.**

**Rating:** 4
**Confidence:** 4

**Review:**

This paper suggests that some of the 'laws' proposed by geologists and seismologists relating likelihood and magnitude of seismic event to past behaviour may be lacking in descriptive power.  To address this, they propose and then fit a few different neural-network based function approximators to  certain parts of a standard Hawkes process based rate estimator for spatio-temporal earth quake event data.

The premise of the paper may well be true (that popular geological 'laws' are much coarser approximations than we could have given modern estimation techniques), however I am not convinced by the analysis.  I am confused by the leap to neural-network approximators in the first place; skipping altogether a simpler set of model comparisons that could be made: e.g. the swap of an exponential law for the plausible (and often more natural) mixture of power laws (i.e. g(m_e) \propto \sum_i w_i m_e^i), or indeed Gaussian process based function approximators (which are as I understand it popular in the geostatistics community and straightforwardly support output uncertainty estimation, a convolutional interpretation, and the ability to control the function space considered viable).

I am a bit confused by the set of 'generic' convolutions proposed in section 6 and 'DCNN' too: are these still causal in time, i.e. are we smoothing or filtering through time?

Some more nit-picky comments:
* In section 3, I find the use of M and m to be confusing.  I'm not sure whether this notation was deliberate or a typo.
* In figure 1 it would be great to see some uncertainty estimates on this MLP estimator function estimator, even via simple dropout or similar.  It would also be nice to have some commentary on the non smoothness and apparent 0 tangent of g(m) for m=2?
* I am not convinced by likelihood gain on validation data I am afraid, particularly for a discretised point process; depending on how the data is distributed these things can be notoriously sensitive to bin size and position, and I don't see much discussion of this in the paper.

---

### Official Review · AnonReviewer2 · 2018-03-12
**Interesting application area of DNN where interpretability is important, the data limited and where even small improvements in understanding/prediction is highly valuable.**

**Rating:** 7
**Confidence:** 3

**Review:**

Prediction of earthquakes is an important problem for which statistical models with few but well-understood parameters have traditionally been used. The authors investigate the potential of deep learning within this context, where the integration is intended to keep the interpretability of the induced representation. The approach is compared with several baselines and provide potential insights on the merit of investigating how the conventional models can be improved. The proposed approach is shown to have a higher likelihood gain than the baselines.

The problem is interesting and the authors does a good job at introducing the reader to the relevant aspects of the field.
The approach seem novel and significant enough for acceptance. The quality of the paper is high.

The authors are well aware of the disadvantage of DNNs within the applied problem and are up front with the restrictions imposed on their experimental results. Their work have the potential to open up further inqueries regarding both the application of DNN in seismic forcasting and the improvement of the conventional models within the same field. The problems the authors face when applying a DNN in this setting are very important problems (small data set, want interpretability, important that the predictions are correct etc.) and work tangenting these problems is merited.

---

### Decision · Program_Chairs · 2018-03-20
**ICLR 2018 Workshop Acceptance Decision**

**Decision:**

Reject

**Comment:**

Based on the reviews, this paper has not been accepted for presentation at the ICLR workshop. However, the conversation and updates can continue to appear here on OpenReview.